# Development and Characterization of a Novel Lineage of Renal Progenitor Cells for Potential Use in Feline Chronic Kidney Disease: A Preliminary Study

**DOI:** 10.3390/cells14171395

**Published:** 2025-09-06

**Authors:** Lara Carolina Mario, Juliana de Paula Nhanharelli, Jéssica Borghesi, Rafaela Rodrigues Ribeiro, Hianka Jasmyne Costa de Carvalho, Thamires Santos da Silva, Mariano del Sol, Rodrigo da Silva Nunes Barreto, Sandra Maria Barbalho, Maria Angelica Miglino

**Affiliations:** 1Department of Anatomy of Domestic and Wild Animals, Faculty of Veterinary Medicine and Animal Science, Universidade de São Paulo (FMVZ/USP), São Paulo 05508-270, SP, Brazil; laracarolina@alumni.usp.br (L.C.M.); jnhanharelli@gmail.com (J.d.P.N.); jehborghesi@hotmail.com (J.B.); rafaelarodriguesribeiro@usp.br (R.R.R.); hiankacarvalho@usp.br (H.J.C.d.C.); thamiresssilva@usp.br (T.S.d.S.); 2Center of Excellence in Morphological and Surgical Studies (CEMyQ), Universidad de La Frontera, Temuco 1145, Chile; mariano.delsol@ufrontera.cl; 3Department of Animal Morphology and Physiology, Faculty of Agricultural and Veterinary Sciences, Universidade Estadual Paulista (UNESP), Jaboticabal 14884-900, SP, Brazil; rodrigo.barreto@unesp.br; 4Department of Biochemistry and Pharmacology, School of Medicine, Universidade de Marília (UNIMAR), Marília 17525-902, SP, Brazil; sandra.barbalho@fatec.sp.gov.br; 5Laboratory for Systematic Investigations of Diseases, Department of Biochemistry and Pharmacology, School of Medicine, Universidade de Marília (UNIMAR), Marília 17525-902, SP, Brazil; 6Department of Structural and Functional Interactions in Rehabilitation, School of Medicine, Universidade de Marília (UNIMAR), Marília 17525-902, SP, Brazil; 7Department of Research, Research Coordination Center, Universidade de Marília (UNIMAR) Charitable Hospital, Universidade de Marília (UNIMAR), Marília 17525-902, SP, Brazil; 8Department of Animal Anatomy, School of Veterinary Medicine, Universidade de Marília (UNIMAR), Marília 17525-902, SP, Brazil; 9Department of Animal Health, Production and Environment, School of Veterinary Medicine, Universidade de Marília (UNIMAR), Marília 17525-902, SP, Brazil

**Keywords:** chronic kidney disease, feline chronic kidney disease, cell therapy, renal progenitor cells, feline, cat health

## Abstract

Chronic kidney disease (CKD) is a common and serious condition in felines. Accordingly, several cell therapies have been studied over the past decades for effective treatments. This study aimed to develop a new lineage of renal progenitor cells for use in cats with CKD. Metanephric and mesonephric progenitor cells were obtained from mesonephros and metanephros tissues of feline conceptuses at four distinct gestational stages. The cultured cells were characterized by their morphology, tumorigenic potential, immunophenotype determined by flow cytometry, and differentiation potential. We then conducted a pilot study in CKD-affected cats, comparing intraperitoneal injections of cultured metanephric progenitor cells (*n* = 4) to a placebo solution (*n* = 3). All four cell types exhibited adhesion and colony formation, but showed no tumorigenic potential. Cells tested positive for renal progenitor markers (CD117, Nephron, and WT1), confirming their identity. Treated cats showed no statistically significant differences (*p* ≤ 0.05) in any of the data analyzed. However, caregivers reported a voluntary increase in appetite after cell administration. Veterinarians confirmed this information during double-blind evaluations conducted after treatment. Although this data are qualitative, no clinical deterioration was observed in cats. Our results suggest that this new lineage of renal progenitor cells did not induce immediate adverse effects, thus supporting its potential for use in cell-based therapies. However, further studies are needed to evaluate its efficacy in treating renal diseases.

## 1. Introduction

Acute and chronic renal diseases are the prevalent causes of mortality in domestic and wild felines [1,2,3]. These conditions impair kidney function, causing tissue injury and reduced blood filtration. Physiological and ultrastructural changes, progressive loss of nephrons, and a significant reduction in glomerular filtration rates characterize chronic kidney disease (CKD). Affected cats often exhibit anorexia, polyuria, polydipsia, and weight loss, which severely impact their overall health [4,5].

Current treatments for CKD are primarily palliative, focusing on improving quality of life and slowing disease progression. These include electrolyte replacement, dietary modifications, and fluid therapy [6,7,8,9]. Although kidney transplantation offers a potential cure, this approach is limited by high costs [4,6]. Recently, alternative treatments combining bioengineering techniques and cell therapy have been explored for CKD treatment [10,11,12,13,14,15]. Although these approaches are still experimental, they show promise as alternatives to current treatments [2,4,7,10].

Mesenchymal stem cells (MSCs), derived from autologous or allogeneic sources, have been widely explored as experimental treatments for CKD. Although promising results have been observed in mice [11,12,13], outcomes in feline patients have been inconsistent, even among individuals treated with identical protocols [2,4,16,17,18,19,20,21]. This inconsistency may arise from various factors, such as cell injection route, donor tissue characteristics, disease duration, and the regulation of MSC differentiation. To address these limitations, we propose using renal progenitor cells (mesonephric and metanephric cells), which may provide better therapeutic outcomes than traditional MSC-based treatments [22,23]. These cells, collected during renal development, are naturally programmed to form renal structures. They have gained attention in regenerative therapies due to their ability to differentiate into multiple renal cell types, secrete cytokines and growth factors, and modulate immune responses. These attributes contribute significantly to renal tissue regeneration [24,25,26,27].

The route of cell administration poses a challenge for this investigation. Intravenous injections, for instance, have been associated with inflammatory reactions, cell clumping, and thromboembolism [11,21]. In contrast, intraperitoneal administration allows for the direct infusion of larger cell volumes into the kidneys. This is because the highly permeable peritoneum facilitates systemic absorption of cells, thereby minimizing the risk of thrombus formation [1,22,28]. However, it is essential to note that the peritoneal cavity interacts dynamically with the immune system through fluid and cell exchange with lymphatic vessels and circulating blood [29].

Given the high mortality rate associated with CKD in felines and the inconsistent outcomes of current cell-based therapies, it is essential to investigate new cellular alternatives, such as renal progenitor cells, and determine more effective administration routes [7,11,24]. In this study, we conducted a pilot evaluation of renal progenitor cells to assess their immediate safety and potential side effects following their application, demonstrating their apparent suitability for treating renal disease. We also aimed to determine an effective route of administration for this therapeutic approach.

## 2. Materials and Methods

### 2.1. Ethical Statement

This study was approved by the Ethics Committee for the Use of Animals of the Faculty of Veterinary Medicine and Animal Science of the University of São Paulo (Protocol No. 9798110215, ID 005608), issued on 17 April 2019. Embryonic and fetal samples were obtained from female cats undergoing ovariohysterectomy during spay/neuter campaigns in São Paulo. The anesthetic protocol included the administration of ketamine, (Basso Pancotte) Nova Alvorada, RS, Brazil; midazolam, (VetSmart) São Paulo, SP, Brazil; xylazine, (Basso Pancotte) Nova Alvorada, RS, Brazil and morphine (VetSmart) São Paulo, SP, Brazil [30,31]. After the uterus was removed, potassium chloride was injected into the uterine artery to ensure the ethical cessation of embryonic and fetal vital signs. The dosage was calculated based on the female’s body weight (1–2 mmol/kg) [32,33,34,35]. Following euthanasia, the animals were transported to the laboratory at the School of Veterinary Medicine and Animal Science, University of São Paulo, where all subsequent procedures were conducted.

### 2.2. Acquisition and In Vitro Culture of Mesonephric and Metanephric Cells

Mesonephric and metanephric cells were collected from embryos and fetuses of 12 pregnant cats (Table 1). The embryos and fetuses were dissected and categorized into four groups based on crown-rump (CR) length (craniocaudal measurement) and external characteristics, as standardized for the species by Evans and Sack [36]. Before culture, the collected kidney tissue was histologically processed and stained with hematoxylin and eosin, and then subjected to immunohistochemistry for CD90, PCNA, caspase-3, and isolectin to confirm its characteristic features. The embryos were classified into early, intermediate, and late gestational stages and organized in the table according to the renal phase observed during development. For a more detailed analysis of renal development, refer to our article on renal development in cats [37].

### 2.3. Cell Isolation and Culture

Embryonic and fetal kidney tissue samples from the mesonephros and metanephros were placed in Petri dishes Bromis, São Paulo, SP, Brazil, and washed five times with phosphate-buffered saline (PBS), Bromis, São Paulo, SP, Brazil, supplemented with 0.5% amikacin Prolab, São Paulo, SP, Brazil, to eliminate potential contaminants. The tissue was mechanically separated, and the explants were transferred to 100 mm culture plates (Corning, Cat. 3296, NY, USA). The plates were filled with one of three culture media: (i) Dulbecco’s Modified Eagle Medium (DMEM), Bromis, São Paulo, SP, Brazil, high glucose, (ii) DMEM-Ham’s F12, Bromis, São Paulo, SP, Brazil, (1:1), or (iii) alpha Minimal Essential Medium (MEM), Bromis, São Paulo, SP, Brazil, each supplemented with 10% fetal bovine serum (FBS), Bromis, São Paulo, SP, Brazil, 0.5% amikacin, and 1% non-essential amino acids. The cell culture protocol was based on that of Dominici et al. [19].

### 2.4. Morphological Analysis of Metanephric and Mesonephric Cultures

Cultured samples were evaluated every three days using a Nikon Eclipse TS-100 microscope, Spctrum, São Paulo, SP, Brazil, to observe and describe morphological variations during growth. For ultrastructural analysis, cells were fixed with 1% osmium tetroxide, Prolab, São Paulo, SP, Brazil, and examined using scanning electron microscopy (SEM).

### 2.5. Cell Viability Test and Cryopreservation

Cultured cells were trypsinized (LGC Technologia, SP, Cat. BR30045-01), stained with trypan blue, Prolab, São Paulo, SP, Brazil, and counted as previously described. Next, they were resuspended in 1.0 mL of freezing solution containing 90% BFS, Bromis, São Paulo, SP, Brazil, and dimethyl sulfoxide (DMSO; LGC Biotecnologia, Cat. 13-0091.01, Cotia, SP, Brazil) and evenly distributed into cryotubes (Cat. 430055, Life Sciences, Corning, NY, USA). The cryotubes were placed in a Mister Frozen device, Thermo Fisher Scientific, São Paulo, SP, Brazil, and stored at –80 °C overnight. Then, they were stored in liquid nitrogen. After one week, the cryotubes were thawed in a 37 °C water bath. Cells were re-stained with trypan blue, and viable cells were counted.

### 2.6. Cell Metabolism Evaluation

Cell metabolism was assessed using the 3-(4,5-dimethylthiazol-2-yl)-2,5-diphenyl tetrazolium bromide (MTT) colorimetric assay [38]. Cells were seeded in 96-well plates (200 μL per well), Bromis, São Paulo, SP, Brazil, and harvested after 24 h for three consecutive days. Subsequently, 10 μL of MTT, Prolab, São Paulo, SP, Brazil, solution was added to each well, and the plates were incubated in a dark chamber at 37 °C for 2 h. After incubation, 100 μL of DMSO was added to each well. After one hour, absorbance was measured at 490 nm using a spectrophotometer.

### 2.7. Cell Characterization by Flow Cytometry

Cells were characterized by flow cytometry following Borghesi’s protocol [38]. For each sample, 10,000 events were counted. The antibodies used to detect renal progenitor cells are listed in Table 2.

### 2.8. Differentiation of Cultured Metanephric and Mesonephric Cells into Adipogenic, Osteogenic, and Chondrogenic Cells

To assess the differentiation potential of the cultured cells, samples were seeded in 24-well plates containing culture medium and divided into four groups: control (1 × 10^3^ cells/mL), adipogenic, osteogenic, and chondrogenic (1 × 10^4^ cells/mL). Upon reaching 80% confluency, the culture medium was replaced with 1 mL of specific differentiation media. The control group received alpha MEM, Bromis, São Paulo, SP, Brazil, supplemented with 75% FBS and 1% penicillin/streptomycin, Prolab, São Paulo, SP, Brazil. After the culture period, cells were stained with hematoxylin and eosin for light microscopy analysis. The osteogenic group was cultured with the StemPro™ Osteogenesis Differentiation Kit (Invitrogen, Carlsbad, CA, USA), and differentiated cells were analyzed using light microscopy after von Kossa and alizarin red staining, Prolab, São Paulo, SP, Brazil. The adipogenic group was cultured in AdipoMAX Differentiation Medium (Sigma-Aldrich, St. Louis, MO, USA), and the differentiated cells were stained with oil red stain (Sigma-Aldrich, O0625), São Paulo, SP, Brazil, for light microscopy analysis. For the chondrogenic group, 2 × 10^6^ cells were centrifuged in a 15 mL tube, Bromis, São Paulo, SP, Brazil, and the resulting pellet was cultured in ChondroMAX Differentiation Medium (Sigma-Aldrich, St. Louis, MO, USA). After differentiation, cells were stained with hematoxylin and eosin for microscopic analysis.

### 2.9. Differentiation of Mesonephric and Metanephric Cells into Endothelial Cells

To induce endothelial differentiation, 1 × 10^4^ mesonephric and metanephric cells were seeded in 24-well plates and cultured for two weeks in CC-3202 medium (Lonza, Salto, SP, Brazil) supplemented with EGM^TM^-2 MV Microvascular Endothelial Cell Growth Medium SingleQuots^TM^ (CC-4147, Lonza, Salto, SP, Brazil). After this period, cells were washed with PBS and fixed in 4% paraformaldehyde, Prolab, São Paulo, SP, Brazil for 30 min. Immunocytochemical labeling for endothelial markers was performed using primary antibodies against endothelial nitric oxide synthase (eNOS, BD610296, BD Biosciences), São Paulo, SP, Brazil, isolectin (I21411, Thermo Fisher Scientific) São Paulo, SP, Brazil, and vascular endothelial growth factor (VEGF, 7422-RBM5-P0, Thermo Fisher Scientific). Primary antibodies were diluted 1:100 and incubated for 2 h in a dark chamber.

To block nonspecific binding, cells were treated with 5% FBS before incubation with secondary antibodies (anti-mouse IgG, SAB4600035, Sigma-Aldrich; anti-rabbit IgG, A21206, Thermo Fisher Scientific) at a 1:500 dilution for 30 min in the dark chamber. Next, cells were washed with Tris-buffered saline, Prolab, São Paulo, SP, Brazil, stained with DAPI (ABCYS, Paris, France), and examined under a fluorescence microscope (LSM 510, Carl Zeiss Microscopy, Jena, Germany).

### 2.10. Evaluation of the Tumorigenic Potential of Cultured Mesonephric and Metanephric Cells

The tumorigenic potential of cultured mesonephric (at 20th and 28th days) and metanephric (at 28th and 50th days) cells was assessed through in vivo experiments. Cells were trypsinized, centrifuged at 1200 rpm for 5 min, and washed with PBS. Next, they were resuspended in 50 µL of saline solution, Prolab, São Paulo, SP, Brazil and injected subcutaneously (1 × 10^6^ cells) into the dorsal region of four Balb/c nude mice using an insulin syringe, Bromis, São Paulo, SP, Brazil. Tumor development was monitored weekly for two months. After a standardized euthanasia procedure, samples from the liver, lungs, kidneys, heart, and spleen were collected and processed for hematoxylin and eosin staining.

### 2.11. Feline CKD Treatment Using Metanephric Cells

The ethics committee approved the experimental treatment protocol. The study involved seven privately owned male and female cats, aged 6–9 years, diagnosed with CKD and undergoing treatment for at least one year. All cats were included only after obtaining informed consent from their owners. The cats were diagnosed with CKD stages I, II, and III according to the International Renal Interest Society (IRIS) criteria [39]. Their blood creatinine levels ranged from 1.6 to 5.0 mg/dL.

Before the experiment, a comprehensive evaluation was conducted, including biochemical profiles, urinalysis, blood pressure measurements, urine protein-to-creatinine ratio (UPC), and serological tests for Feline Immunodeficiency Virus (FIV) and Feline Leukemia Virus (FeLV). All cats tested negative for FIV and FeLV and showed no evidence of concurrent diseases, acute renal injury, urolithiasis, urinary obstructions, masses, or kidney abnormalities (e.g., polycystic kidney disease).

The cats were divided into a control group (*n* = 3; one cat per CKD stage I, II, and III) and an experimental group (*n* = 4; two cats in stage I, one in stage II, and one in stage III). Detailed characteristics of the animals, including disease staging according to the IRIS classification, are provided in Table 3.

### 2.12. Intraperitoneal Cell Injection for Cell Therapy

The experimental group (*n* = 4) received 2 × 10^6^ metanephric cells resuspended in 3 mL of saline solution, while the control group (*n* = 3) received a PBS placebo to ensure a double-blinded study design. Cell injection was performed with the cat in dorsal recumbency under manual restraint. A mid-abdominal trichotomy was conducted, followed by antisepsis with 0.5% alcoholic chlorhexidine, Prolab, São Paulo, SP, Brazil. A 24G catheter was inserted under ultrasound guidance in the mesogastric region, 2 cm below the umbilical scar, along the linea alba. The catheter was fully inserted, and the stylet was removed. Ultrasound imaging confirmed that no abdominal structures were punctured during the procedure.

### 2.13. Laboratory Monitoring and Clinical Evaluation

Vital parameters (blood pressure, heart rate, body temperature, and respiratory rate) were measured immediately after injection. The cats were monitored for 20 min to detect any signs of pain or discomfort during abdominal palpation. Hemograms, serum urea, creatinine, phosphorus, symmetric dimethylarginine (SDMA), and urinalysis were conducted at three time points: before injection (day 0), and on days 7 and 14 post-injection. On days 7 and 14, caretakers were queried about the cats’ behavior, including water and food intake, as well as the volume of urine and feces excreted. Physical examinations were also conducted, recording body weight, body condition score, hydration levels, oral mucosa color, and vital parameters.

### 2.14. Statistical Analysis

Statistical analysis was performed using the Scheirer–Ray–Hare test for non-parametric data. The *t*-test was used to compare the experimental and control groups at different time intervals (0, 7, and 14 days) and to calculate overall averages between the two groups. Differences were considered statistically significant at *p* ≤ 0.05. All data and statistical results are presented in Table 4.

## 3. Results

### 3.1. Culture, Differentiation, Characterization, and Assessment of Tumorigenic Potential of Mesonephric and Metanephric Cells

Macroscopically, the mesonephros was elongated during early gestation but regressed after the onset of metanephric development in mid-gestation. At this stage, the metanephros lacked cortical or medullary division, which was completed only during late gestation (Figure 1).

Cultured cells exhibited two distinct morphologies: (i) a fusiform shape with centralized nuclei and elongated cytoplasm, and (ii) a cuboidal shape with centralized nuclei forming flat colonies on the culture plates (Figure 2A–D). Metabolic activity, assessed via the MTT assay, was highest in alpha MEM, declining after 48 h. In contrast, cells cultured in DMEM high glucose and DMEM-Ham’s F12 exhibited lower growth rates (Table 5). These findings were further supported by higher optical density (O.D.) values in alpha MEM compared to other media (Figure 2). Likewise, cells cultured in alpha-MEM medium supplemented with FBS showed a fibroblastic shape, adhered well to the plate, and reached 50% confluence. Cellular proliferation peaked 48 h post-thawing, and cells retained their morphological characteristics and growth patterns after cryopreservation (Figure 2).

Cells remained viable after thawing, retaining their original characteristics. Mesonephric cells from early gestation had a viability of 93%, whereas those from mid-gestation had a viability of 92%. In contrast, metanephric cells from mid-gestation had a viability of 85%, whereas cells from late gestation had a viability of 92%. Flow cytometry revealed differential expression levels of PCNA, CD34, CD105, CD90, MHC I, MHC II, CD117, Nephrin, WT1, and CD117 plus Nephrin across different gestational periods (Figure 3; Table 6).

Evaluation of tumorigenic potential in mesonephric (20th and 28th days) and metanephric (28th and 50th days) cells showed no tumor development in the Balb/c nude mice. Histological analysis of lung, heart, liver, and kidney samples confirmed no significant abnormalities (Figure 4).

In adipogenic differentiation medium, cells developed dispersed adipose vesicles and other morphological changes. The differentiated cells were elongated, and their nuclei migrated to the periphery, contrasting with the fibroblastic morphology of control cells (Figure 5A–D). In osteogenic differentiation medium, the cells developed ramifications and subsequently formed calcification points in the extracellular matrix (Figure 5E–H). Following chondrogenic differentiation, the cells assumed a round, chondrocyte-like morphology, distinct from that of control cells (Figure 5I–L). At the end of endothelial differentiation, mesonephric (20th and 28th days) and metanephric (28th and 50th days) cells stained positively for isolectin, VEGF, and eNOS antibodies (Figure 6).

### 3.2. Cellular Application—Pilot Study

The treated cats showed no side effects following cell administration. Notably, they showed an increase in food intake. Regarding quantitative parameters, no statistically significant changes were observed in body weight or leukocyte counts (*p* ≤ 0.06). None of the animals in either group exhibited clinical signs of infection, fever, or changes in leukocyte morphology. Creatinine and phosphorus levels remained unchanged in both groups (*p* ≤ 0.05). Similarly, SDMA showed no statistically significant changes between the treated and control groups (Figure 7).

Qualitative clinical observations, including hydration and appetite disturbances, were reported by owners to the veterinarians conducting the double-blind tests. These observations, recorded before and after the injections, were compiled and are presented in Table 7.

## 4. Discussion

The development of new treatments for feline CKD is often hindered by small sample sizes and variability in clinical signs among affected cats, which precludes the implementation of double-blind studies and long-term evaluations [21,40]. Although stromal stem cells derived from adipose, uterine, amniotic, and fibroblastic tissues have been explored in previous studies, their benefits have been limited to improving quality of life rather than offering curative outcomes [2,4,7,11,17,18,21]. Instead, this study aimed to evaluate a cell therapy approach using metanephric progenitor cells. Given the capacity of the metanephros to develop into definitive kidneys, we drew upon recent investigations of renal progenitor cells as effective treatments for renal injury. Importantly, when tested in vivo, these cells demonstrated the ability to integrate and reproduce nephron epithelial components following transplantation [30,32,33,41,42,43].

Because the cells were obtained at four different gestational periods (mesonephros at 20 and 28 days, and metanephros at 28 and 50 days), they initially exhibited distinct morphological characteristics; however, after culture, they evolved into fusiform cells with centralized nuclei and elongated cytoplasm [44,45].

Immunophenotypic analysis showed that mesonephric cells from early gestation displayed higher proliferation rates than those from day 28. This difference may be attributed to mesonephric regression during this period [46,47]. In contrast, metanephric cells from both gestational periods exhibited similar expression of PCNA, a marker of cellular proliferation. These results were expected, as the mesonephros develops into the kidneys during the neonatal period while retaining its capacity to proliferate and differentiate into various cell types [48].

The cultured cells expressed low levels of CD34, a marker for hematopoietic and endothelial cells [49]. Mesenchymal differentiation was confirmed by the presence of CD90 and CD105, which were expressed throughout all gestational periods. CD90 is highly expressed in thymocytes, mesenchymal cells, fibroblasts, and endothelial cells, whereas CD105 (endoglin) is involved in transforming growth factor β signaling, which regulates vascular proliferation, development, remodeling, cell proliferation, and embryonic development [19,50,51].

The presence of CD117, Nephrin, and WT1 confirms the potential of these cells to differentiate into renal cells, as these markers are specific to proteins found in renal tissue. WT1 plays an essential role in the development of renal progenitor cells [27,52,53,54,55]. Moreover, the high expression of CD117 is of utmost importance, as this protein, known as c-kit, identifies a critical type of renal precursor cell. C-kit cells, which resemble stem cells, help improve renal function after acute ischemic injury in rats [56,57]. Nephrin is a key protein for podocytes, mediating critical cell signaling pathways and maintaining the normal function of the glomerular filtration unit in the kidneys. Reductions in nephrin levels are often associated with various renal diseases [52,58,59]. In this context, WT1 is vital for the proliferation of renal progenitor cells [53]; it directly influences the development of embryonic nephrons and promotes the regeneration of renal proximal tubules after acute kidney injury [55,60]. Other markers, such as Six2 and Cited1, are well established for identifying renal progenitor cells in humans and mice [61,62,63,64], but are not yet available for feline cells, which limits the expansion of cellular labeling in these animals. Although gene expression profiling could represent a valuable alternative, especially given the limited commercial availability of feline-specific antibodies, such data are also scarce for this species. As highlighted by Lindström et al. [65], most gene expression datasets about humans and mice indicate that feline-specific validation would still be required. Consequently, due to these technical issues, this study was unable to include additional molecular features.

The non-significant expression of major histocompatibility complex proteins (MHC I and MHC II) in all four cell types supports their association with the body’s defenses [66,67]. To assess the safety of progenitor kidney cells and the potential for tumor development, these cells were injected into Balb/c nude mice, and no tumor formation was observed [68]. Additionally, to assess their differentiation potential, the cells were cultured in induction media for adipogenic, osteogenic, and chondrogenic lineages [19]. Renal progenitor cell lineages showed differentiation potential into endothelial lineages, as evidenced by positive staining for isolectin, VEGF, and eNOS. The ability of these cells to differentiate into endothelial lineages was also observed, supporting renal vasculogenesis during embryonic development [69].

The use of renal progenitor cells in the experimental group was compared to the application of PBS in the control group. Although several studies have used mesenchymal cells to treat kidney diseases [24,25,26,27], we chose not to include this cell lineage as a control. Our goal was to evaluate the effects of renal progenitor cells exclusively, without the interference of the paracrine known impact of mesenchymal cells. In this way, we directly assessed the interactions and potential side effects of this novel cell lineage, ensuring that the action of mesenchymal cells did not influence the observed interactions.

Treated cats showed no immediate side effects following the administration of renal progenitor cells. The intraperitoneal route allowed for the infusion of larger volumes of cell therapy, facilitating systemic absorption and minimizing the risk of thrombus formation. Our study was based on the application of this cell lineage as outlined by Barros et al. [70,71] who demonstrated in a rat model that intraperitoneally injected stem cells migrated to the kidney, where their paracrine effects promoted the recovery of damaged cells. Moreover, this region is poorly innervated, which reduces discomfort for the animal and eliminates the need for analgesics or sedatives [72,73,74,75].

No statistically significant differences in body weight were observed, though treated cats showed increased appetite compared to the control group. Although these changes were not statistically significant (*p* ≤ 0.005), the voluntary increase in food intake suggests a positive response. Improvements in quality of life are essential outcomes and align with the Core Outcome Set for assessing treatment efficacy in feline CKD trials [39,40,72]. SDMA is an early marker of CKD and correlates with glomerular filtration rate [60,76]. In this research, SDMA showed no statistically significant differences (*p* ≤ 0.005) between the treated and control groups. Due to the small sample size and heterogeneous renal disease stages in the treated groups, this study primarily demonstrated the safety of renal progenitor cells administration, with no side effects after application. However, further studies with larger sample sizes and more homogeneous groups of CKD should be conducted to confirm these findings. The small sample size was primarily due to challenges in guardian compliance, as many guardians did not return for follow-up visits, thereby compromising study continuity. One of the main challenges in research involving guardian-owned animals is raising awareness about the importance of initiating and completing treatment so that the data obtained can significantly contribute to the development of new therapies [76,77,78].

## 5. Conclusions

Cell therapy studies are fundamental for advancing treatments for chronic diseases. Progenitor cells offer greater therapeutic potential due to their pre-differentiated state in the source tissue. This study shows that metanephric progenitor cells from late gestational stages are effective, as they express higher levels of renal protein markers. This finding makes metanephric progenitor cells ideal candidates for therapeutic applications. Furthermore, the administration of this novel progenitor cell lineage showed no side effects, indicating it is safe for use in cats with renal disease. However, further studies with larger sample sizes are needed to validate these findings fully.

## Figures and Tables

**Figure 1 cells-14-01395-f001:**
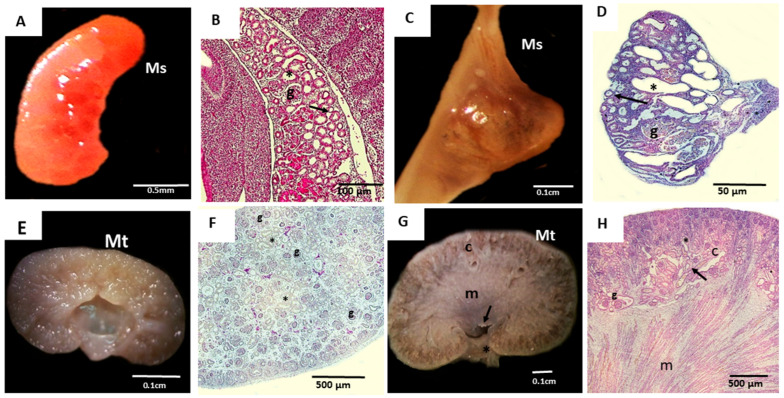
Macroscopic and microscopic analysis of the metanephros (Mt) and mesonephros (Ms) from domestic cats. (**A**)—The elongated mesonephros at 19 days of gestation; (**B**)—histological section of the 19-day mesonephros displaying glomeruli (g), mesonephric ducts (arrow), and mesonephric tubules (*); (**C**)—mesonephros at 28 days, showing a more oval shape; (**D**)—histological section of the 28-day mesonephros, with glomeruli (g), mesonephric ducts (arrow), and mesonephric tubules (*); (**E**)—metanephros at 28 days, lacking corticomedullary division; (**F**)—histological section of the 28-day metanephros, showing glomeruli (g) and tubules (*) dispersed throughout the tissue; (**G**)—metanephros at 50 days of gestation, showing all macroscopic divisions, including the cortical region (c), medullary region (m), renal crest (arrow), and renal pelvis (*); (**H**)—histological section of the 50-day metanephros, showing the cortical region (c), with glomeruli (g), proximal convoluted tubules (arrow), distal convoluted tubules (*), and a medullary region (m), where the tubular structures are evident.

**Figure 2 cells-14-01395-f002:**
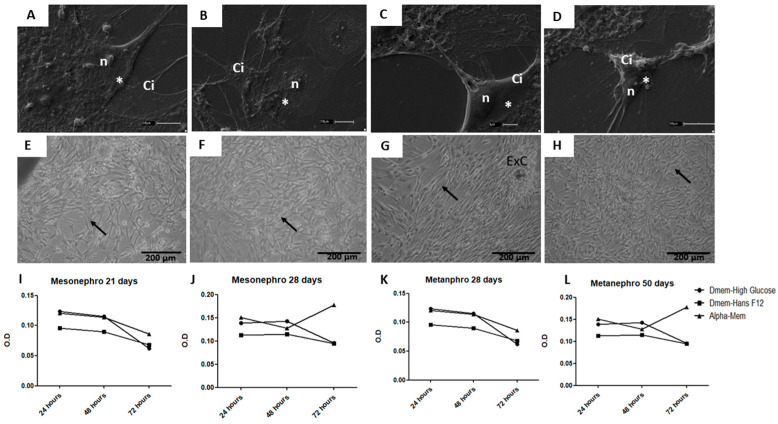
Cell morphology of mesonephric (20th and 28th days) and metanephric (28th and 50th days) conceptuses, and proliferation in three culture media (alpha MEM, DMEM high glucose, and DMEM-Ham’s F12). (**A**,**B**)—Cells at 20 days (**A**) and 28 days (**B**), displaying fusiform (triangle) and cuboidal (*) shapes, with a centralized mesonephric nucleus (n) surrounded by cytoplasm (Ci); (**C**,**D**)—metanephric cells at 28 days (**C**) and 50 days (**D**), displaying fusiform (triangle) and cuboidal (*) shapes, with a centralized mesonephric nucleus (n) surrounded by cytoplasm (Ci); (**E**–**H**)—Mesonephric and metanephric cells at 20, 28, and 50 days showing colony formation, adherence to the culture plate (arrow); and cellular explant (ExC), representing a tissue fragment cultured following mechanical dissociation. (**I**–**L**)—culture of mesonephric cells at 21 and 28 days and metanephric cells at 28 and 50 days, showing cellular proliferation and peak growth after 48 h of culture.

**Figure 3 cells-14-01395-f003:**
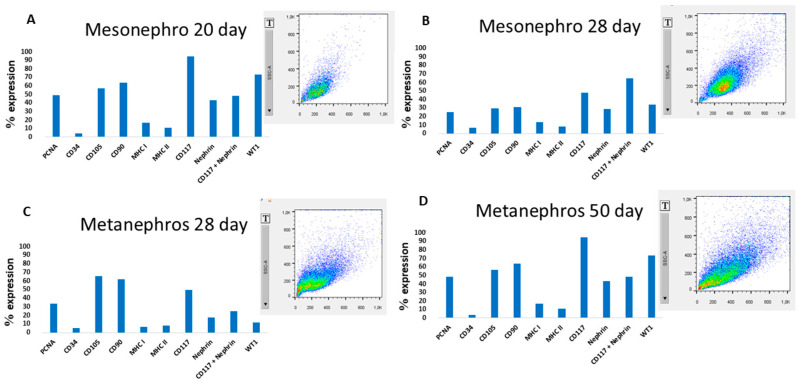
Flow cytometry analysis of protein expression in cultured. The dot plot colors indicate cell density (red/orange = high, green = intermediate, blue = low). In (**A**), mesonephric cells at 20 days, In (**B**), mesonephric cells at 20 days. In (**C**), metanephric cells at 28 days. In (**D**), metanephric cells at 50 days of gestation.

**Figure 4 cells-14-01395-f004:**
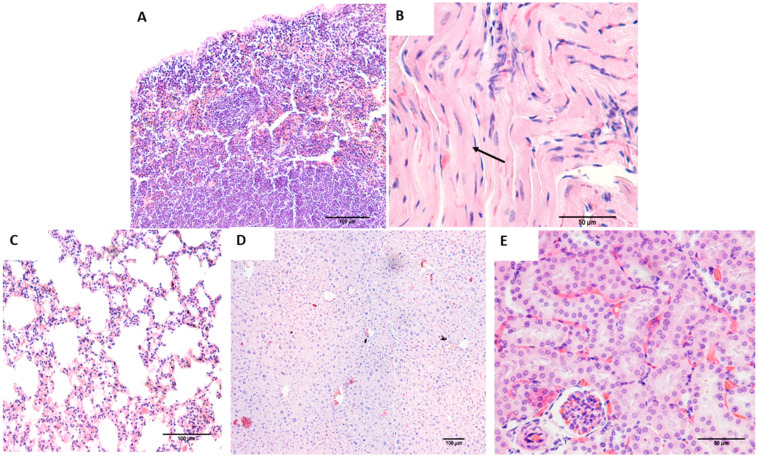
Representative evaluation of the tumorigenic potential of 20- and 28-day mesonephric cells and 28- and 50-day metanephric cells in Balb/c nude mice. (**A**)—Histology of the lung; (**B**)—histology of the heart, evidencing myofibers (arrow); (**C**)—histology of the spleen; (**D**)—histology of the liver; (**E**)—histology of the kidney.

**Figure 5 cells-14-01395-f005:**
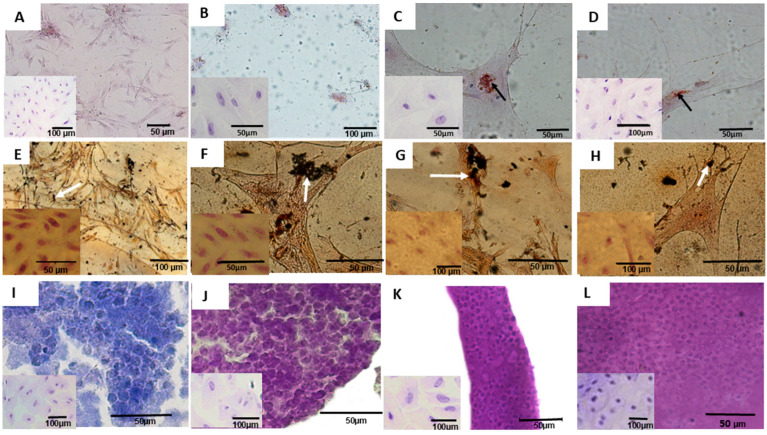
Differentiation of metanephric (20th and 28th days) and mesonephric (24th and 50th days) cells into adipogenic, osteogenic, and chondrogenic cells. (**A**–**D**)—Adipogenic differentiation of cultured cells, with adipose vesicles (arrows), dispersed morphology, and rounded shapes with nuclei located at the periphery of the cytoplasm; (**E**–**H**)—osteogenic differentiation, with calcification points (arrows) and diffuse morphology compared to control samples, which exhibited centralized nuclei. Scale bar: 50 µm; (**I**–**L**)—chondrogenic differentiation, featuring rounded cells and control cells (top) with fibroblastic colony shapes. Scale bar: 100 µm.

**Figure 6 cells-14-01395-f006:**
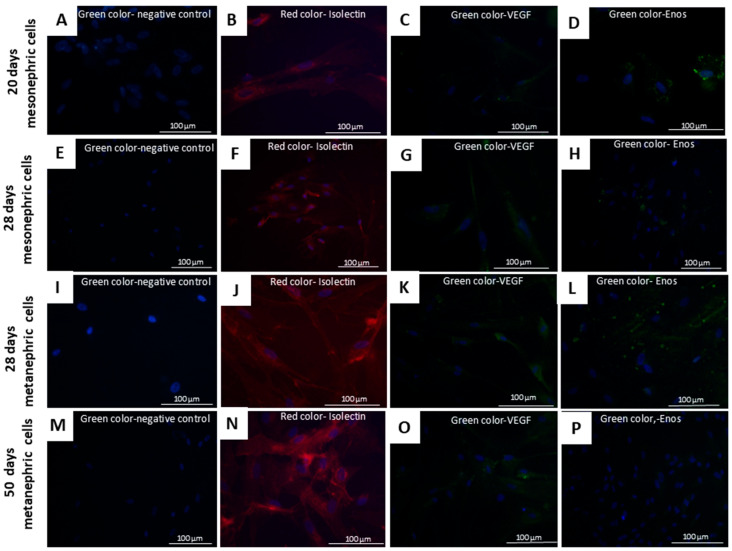
Cell differentiation of mesonephric and metanephric cells into endothelial cells. (**A**)—Negative control; (**B**–**D**)—immunopositive staining for isolectin (**B**), VEGF (**C**), and eNOS (**D**) in mesonephric cells at 20 gestational days; (**E**)—negative control for 28-day mesonephric cells; (**F**–**H**)—immunopositive staining for isolectin (**F**), VEGF (**G**), and eNOS (**H**); (**I**)—negative control for 28-day metanephric cells; (**J**–**L**)—Immunopositive staining for isolectin (**J**), VEGF (**K**), and eNOS (**L**) in metanephric cells at 28 days. (**M**)—negative control for metanephric cells at 50 days; (**N**–**P**)—Immunopositive staining for isolectin (**N**), VEGF (**O**), and eNOS (**P**) in metanephric cells at 50 days. Objective magnification: 400×. Green = VEGF/eNOS, Red = isolectin, Blue = DAPI; negative controls.

**Figure 7 cells-14-01395-f007:**
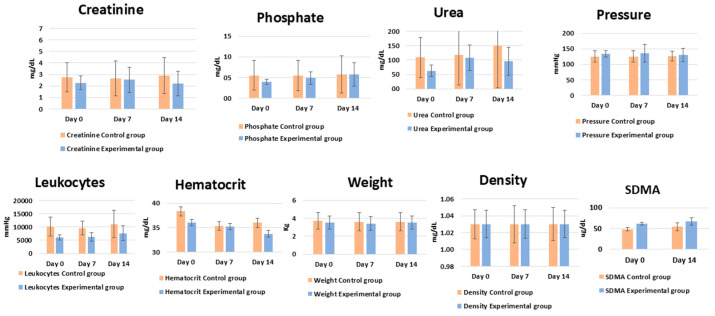
Average clinical and laboratory results for control and experimental groups of cats on days 0, 7, and 14.

**Table 1 cells-14-01395-t001:** Number of mesonephros and metanephros samples collected during each gestational period based on crown-rump (CR).

Cat Mesonephros		Cat Metanephros	
15th to 21st Days of Gestation	22nd to 30th Days of Gestation	24th to 30th Days of Gestation	50th to 60th Days of Gestation
Early gestation	Mid-gestation	Mid-gestation	Late gestation
CR = 0.5 − 2.1 cm	CR = 2.5 − 6.5 cm	CR = 3.1 − 6.5 cm	CR = 9.3 − 13 cm
3 embryos	3 embryos and fetuses	3 fetuses	3 fetuses

**Table 2 cells-14-01395-t002:** Antibodies (and their functions) for characterization of renal progenitor cells via flow cytometry.

Antibody	Catalog Number, Manufacturer	Function
CD34	ab8158, Abcam, Nova Analítica, SP, Brazil	Hematopoietic progenitors
CD117	A4502, Dako, CiteAB, Bath, Somerset, UK	Renal precursor
CD105	ab53321, Abcam, Nova Analítica, SP, Brazil	Endothelial cells
PCNA	sc-46, Santa Cruz Biotechnology, Interprise, Paulinia, SP, Brazil	Cellular proliferation
CD90	WS0809D-100, Kingfisher Biotech, Quimigen, Alverca do Ribatejo, Lisboa, Portugal	Mesenchymal cells
MHC I	18067213, Thermo Fisher Scientific, São Paulo, SP, Brazil	Expression of major histocompatibility complex (MHC I) proteins
MHC II	17092082, Thermo Fisher Scientific, São Paulo, SP, Brazil	Expression of major histocompatibility complex (MHC II) proteins
WT1	ab89901, Abcam Nova Analítica, SP, Brazil	Kidney progenitor cells during development
Nephrin	ab216692, Abcam Nova Analítica, SP, Brazil	Kidney progenitor cells for podocytes

**Table 3 cells-14-01395-t003:** Animals with chronic kidney disease (CKD), where CF—castrated female; CM—castrated male; UB—undefined breed; Group E—experimental (application of renal stem cells 2 × 10^6^); Group C—control.

Anima l	Group	Description	Inclusion Criteria				Internship CKD (IRIS)
			Creatinine Level	DU	Morphological Change	FIV/ FeLV	
1	E	8 years CF/UB	2.25 mg/dL	1.016	Yes		2
2	E	9 years CF/UB	2.89 mg/dL	1.021	Yes		3
3	E	6 years CF/SRD	1.5 mg/dL	1.050	Yes		1
4	E	7 years CM/UB	1.70 mg/dL	1.017	Yes		1
1	C	7 years CF/UB	1.6 mg/dL	1.050	Yes		1
2	C	8 years CF/UB	4.1 mg/dL	1.019	Yes		3
3	C	9 years CM/UB	2.57 mg/dL	1.021	Yes		2

**Table 4 cells-14-01395-t004:** Table with the average values obtained for the animals subjected to treatment compared to those in the control group.

	Experimental Group—Average			Control Group—Average			*t*-Test Day by Day			*t*-Test Control vs. Treatment
	0 days	7 days	14 days	0 days	7 days	14 days	0 days	7 days	14 days	
Creatinine	2.3	2.5	2.2	2.8	2.7	3.2	0.59	0.90	0.38	0.415
Phosphate	4.0	4.9	5.8	5.5	5.5	5.7	0.43	0.78	0.97	0.888
Urea	62.8	107.6	96.0	109.0	116.9	148.8	0.25	0.88	0.52	0.490
Pressure	133.8	136.3	130.0	125.0	125.0	126.7	0.44	0.58	0.82	0.517
Leukocytes	6100.0	6325.0	7675.0	10,200.0	9600.0	11,100.0	0.08	0.09	0.31	0.060
Hematocrit	36.0	35.3	33.8	38.3	35.3	36.0	0.61	0.99	0.75	0.772
Weight	3.5	3.4	3.5	3.7	3.6	3.6	0.76	0.79	0.90	0.750
Density	1.0	1.0	1.0	1.0	1.0	1.0	0.77	0.85	0.86	0.775

**Table 5 cells-14-01395-t005:** Analysis of metabolic activity of different media in distinct cell lineages derived from renal progenitors.

	Mesonephros (21 Days)			Mesonephros (28 Days)			Metanephros (28 Days)			Metanephros (50 Days)		
Hour	DMEM HIGH	DMEM HAM’S F12	ALPHA MEM	DMEM HIGH	DMEM HAM’S F12	ALPHA MEM	DMEM HIGH	DMEM HAM’S F12	ALPHA MEM	DMEM HIGH	DMEM HAM’S F12	ALPHA MEM
24	0.17	0.07	0.13	0.049	0.048	0.048	0.049	0.049	0.064	0.100	0.090	0.120
	0.15	0.07	0.12	0.124	0.120	0.147	0.131	0.085	0.123	0.170	0.080	0.090
	0.14	0.07	0.13	0.140	0.113	0.151	0.129	0.083	0.124	0.110	0.070	0.090
	0.16	0.07	0.19	0.137	0.104	0.134	0.119	0.126	0.125	0.100	0.100	0.130
	0.09	0.06	0.07	0.153	0.153	0.170	0.116	0.088	0.110	0.050	0.070	0.080
48	0.190	0.11	0.10	0.048	0.048	0.049	0.049	0.049	0.048	0.090	0.060	0.150
	0.350	0.09	0.19	0.156	0.101	0.172	0.116	0.082	0.111	0.130	0.100	0.080
	0.360	0.08	0.19	0.158	0.120	0.141	0.136	0.085	0.115	0.070	0.060	0.090
	0.280	0.07	0.27	0.128	0.124	0.159	0.106	0.112	0.112	0.130	0.060	0.100
	0.060	0.05	0.07	0.128	0.114	0.140	0.103	0.081	0.118	0.070	0.050	0.060
72	0.17	0.06	0.11	0.052	0.063	0.055	0.055	0.062	0.052	0.150	0.090	0.370
	0.20	0.06	0.10	0.105	0.141	0.256	0.065	0.064	0.083	0.070	0.080	0.250
	0.17	0.07	0.13	0.091	0.112	0.184	0.061	0.072	0.103	0.310	0.080	0.150
	0.21	0.07	0.12	0.101	0.123	0.157	0.066	0.068	0.075	0.120	0.120	0.240
	0.06	0.05	0.06	0.085	0.104	0.112	0.056	0.069	0.084	0.060	0.050	0.060

**Table 6 cells-14-01395-t006:** Percentage of protein markers in cultured mesonephric and metanephric cells across different gestational periods.

Antibody	Gestational Period			
	Mesonephros (20 Days)	Mesonephros (28 Days)	Metanephros (28 Days)	Metanephros (50 Days)
PCNA	35.7%	25.4%	33.7%	48.8%
CD34	10.9%	6.9%	5.2%	3.9%
CD105	16.4%	29.4%	65.4%	56.7%
CD90	12.7%	30.9%	61.9%	63.6%
MHC I	6.7%	13.7%	6.5%	16.9%
MHC II	9.4%	7.9%	7.9%	10.9%
CD117	30.9%	47.9%	49.5%	94.6%
Nephrin	30.7%	28.9%	17.9%	43.3%
CD117 plus Nephrin	65.1%	65.1%	24.9%	48.5%
WT1	13.7%	34.2%	11.4%	73.2%

**Table 7 cells-14-01395-t007:** Qualitative clinical symptoms associated with hydration and appetite disturbances observed in the experimental group (EG) and control group (CG) before (day 0) and after placebo or intraperitoneal cell injection (day 7 and day 14), observed by veterinarians and information pro-vided by the caretaker based on observations made in the home environment during the study period.

	Day 0		Day 7		Day 14	
Manifestation	EG	CG	EG	CG	EG	CG
Languidness	-	-	-	1 (33.3%)	-	1 (33.3%)
Dehydration	-	-	1 (25%)	-	-	1 (33.3%)
Hyporexia	-	-	1 (25%)	1 (33.3%)	-	-
Polyuria	3 (75%)	2 (66.6%)	3 (75%)	2 (66.6%)	3 (75%)	2 (66.6%)
Polydipsia	3 (75%)	2 (66.6%)	3 (75%)	2 (66.6%)	3 (75%)	2 (66.6%)
Vomit	-	-	-	-	-	1 (33.3%)
Diarrhea	-	-	1 (25%)	-	-	-

## Data Availability

The raw data supporting the conclusions of this article will be made available by the authors on request.

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
