# Peer review of "Development and Characterization of a Novel Lineage of Renal Progenitor Cells for Potential Use in Feline Chronic Kidney Disease: A Preliminary Study"

_cells, 2025, doi:10.3390/cells14171395_

Round 1
Reviewer 1 Report
Comments and Suggestions for Authors
Dear authors,
This study represents an ambitious and pioneering effort to explore a novel form of cell therapy for feline chronic kidney disease (CKD), employing a unique approach centered on the isolation, culture, and evaluation of fetal-derived mesonephric and metanephric progenitor cells. The absence of tumorigenicity, the confirmation of safety, and the assessment of multipotency are significant accomplishments. However, several important limitations and areas for improvement remain. The following reviewer comments are provided for consideration:
- The clinical trial portion of this study involved an extremely limited sample size—only four cats in the treatment group and three in the control group. Such a small cohort leads to markedly low statistical power. Under these conditions, it is highly likely that any therapeutic effects, even if present, could not be reliably detected. Therefore, the lack of statistically significant differences cannot be interpreted as evidence of no effect. This raises a substantial risk of type II error. To generate scientifically reliable results, an a priori power analysis should be conducted to determine an appropriate sample size.
- In this study, the control group received only PBS, without comparison to existing stem cell-based treatments such as adipose-derived mesenchymal stem cells. Consequently, it is impossible to evaluate whether the introduced renal progenitor cells offer superior, equivalent, or inferior therapeutic potential relative to established treatments. For the findings to demonstrate clinical innovation or utility, a minimum requirement would be the inclusion of comparisons with current standard-of-care therapies.
- The study's evaluation of clinical efficacy relies heavily on subjective assessments from cat owners, including observations of increased appetite and improved activity. While these outcomes may reflect meaningful clinical improvement, they are inherently vulnerable to bias, especially given the small sample size. To enhance objectivity and reproducibility, future studies should incorporate quantifiable physiological and biochemical markers, such as longitudinal SDMA levels, glomerular filtration rate, and urinary protein concentrations.
- No statistically significant differences were observed between the treatment and control groups in blood chemistry markers such as creatinine, phosphate, BUN, or SDMA. This is particularly concerning, as SDMA is recognized as an early and sensitive biomarker of renal dysfunction. The lack of change in these quantitative indicators suggests that the cell therapy may not have exerted a measurable effect on CKD pathophysiology. Without demonstrable functional improvement, a claim of safety alone is insufficient to support clinical relevance.
- The study identifies renal progenitor cells based on the expression of markers such as CD117, WT1, and NEPHRIN. However, it does not evaluate widely accepted progenitor cell markers like SIX2 and CITED1, which are standard in human and murine renal research. Even if antibody limitations in cats present a challenge, alternative approaches such as gene expression profiling should have been pursued to strengthen the identification and characterization of the cell population.
- Despite CKD being a progressive, long-term condition, the study's observation period was limited to just 14 days. Such a short timeframe is inadequate for assessing sustained improvements in kidney function or the long-term impact of treatment on disease progression. The appropriateness of the CKD model itself is questionable under these constraints, and a study duration of at least several months would be necessary to evaluate therapeutic durability.
- The study utilized intraperitoneal injection for cell delivery, justifying this choice by citing ease of administration and reduced animal discomfort. However, the scientific rationale for this route is unclear. If renal homing is a desired outcome, intravenous or intra-arterial delivery is more common and supported by prior studies demonstrating superior renal engraftment. The authors also fail to provide evidence that the administered cells reached the kidney after IP injection, leaving the actual mechanism of therapeutic action uncertain.
Author Response
RESPONSE TO REVIEWERS' COMMENTS
Manuscript number: cells-3779232 ― Cells (MDPI)
"Development and Characterization of a Novel Lineage of Renal Progenitor Cells for Potential Use in Feline Chronic Kidney Disease: A Preliminary Study"
The authors of this document wish to express their deepest gratitude to the Editor-in-Chief and the Reviewer for their thorough and insightful evaluation of our manuscript. Their expert feedback has been invaluable in enhancing the quality of our work. We have carefully considered and diligently implemented each suggestion, significantly improving the manuscript. We have made substantial revisions to address the points raised. These noteworthy changes are marked mainly with YELLOW-highlighted text throughout the document for ease of reference. A note will be provided for the referee's attention for corrections highlighted in a different color. Additionally, we have prepared a detailed and comprehensive response to each comment and suggestion. This response is organized in a "point-by-point" format below, ensuring that every concern has been thoroughly addressed and explained. We sincerely appreciate the time and effort invested by the Editor-in-Chief and the Reviewer, and we believe their contributions have significantly strengthened the final version of our manuscript.
Reviewer #1
Comment #1
Dear authors,
This study represents an ambitious and pioneering effort to explore a novel form of cell therapy for feline chronic kidney disease (CKD), employing a unique approach centered on the isolation, culture, and evaluation of fetal-derived mesonephric and metanephric progenitor cells. The absence of tumorigenicity, the confirmation of safety, and the assessment of multipotency are significant accomplishments. However, several important limitations and areas for improvement remain. The following reviewer comments are provided for consideration.
Response:
We sincerely appreciate the valuable observations and suggestions, which have significantly contributed to the improvement of our work. All the reviewer’s suggestions have been carefully analyzed, addressed, and are detailed below.
Comment #2
The clinical trial portion of this study involved an extremely limited sample size—only four cats in the treatment group and three in the control group. Such a small cohort leads to markedly low statistical power. Under these conditions, it is highly likely that any therapeutic effects, even if present, could not be reliably detected. Therefore, the lack of statistically significant differences cannot be interpreted as evidence of no effect. This raises a substantial risk of type II error. To generate scientifically reliable results, an a priori power analysis should be conducted to determine an appropriate sample size.
Response
We appreciate the reviewer’s pertinent observation regarding the sample size used in this study. Indeed, we acknowledge that the limited sample size constrains the statistical robustness, particularly concerning the evaluation of therapeutic efficacy (lines: 462; 463–464; 467–469). However, it is important to clarify that the primary focus of this work was the preliminary safety assessment of fetal renal stem cell administration in cats, rather than the determination of statistically significant clinical outcomes. Therefore, we consider the project to be viable in light of its stated objective.
As highlighted at the end of the Introduction, “the initial aim was to evaluate whether the use of fetal stem cells, derived from renal progenitor cells, could be considered safe in cats with chronic kidney disease at different stages of the condition” (lines 91–93), without inducing clinically or laboratorially relevant adverse effects. Accordingly, the absence of adverse events and the clinical and laboratory stability of the treated animals—despite the small sample size—provide a preliminary indication of the biosafety of these cell lines in a feline model (lines 454–455).
We fully understand, as noted by the reviewer, that the lack of statistical significance should not be interpreted as an absence of therapeutic effect, especially considering the sample size limitation. In fact, we do not propose any conclusions regarding efficacy in this study. Instead, we present initial findings intended to support future stages, which will involve an expanded sample size and an appropriately defined a priori power analysis (lines 469 - 473). These steps are unquestionably essential for validating any potential clinical benefit observed.
Thus, we reiterate that this study was not designed to test efficacy hypotheses with high statistical power, but rather to fulfill a fundamental exploratory step: ensuring that the use of fetal renal stem cells does not result in deleterious effects in feline recipients.
Comment #3
In this study, the control group received only PBS, without comparison to existing stem cell-based treatments such as adipose-derived mesenchymal stem cells. Consequently, it is impossible to evaluate whether the introduced renal progenitor cells offer superior, equivalent, or inferior therapeutic potential relative to established treatments. For the findings to demonstrate clinical innovation or utility, a minimum requirement would be the inclusion of comparisons with current standard-of-care therapies.
Response
We appreciate the reviewer’s observation and acknowledge that comparisons with well-established cell therapies, such as mesenchymal stem cells (MSCs), could provide valuable insights into therapeutic efficacy. However, it is important to emphasize that the primary scope of this preliminary study was not to compare different cell types, but rather to independently assess the safety of fetal renal progenitor cell administration in cats with chronic kidney disease.
The use of PBS as a control was intended to eliminate any paracrine effects associated with other cell types, such as MSCs, thereby ensuring that all observed outcomes could be attributed exclusively to the fetal renal cells (lines 457 - 453). As indicated in the manuscript, “the initial aim was to assess the feasibility and safety of renal progenitor cell lines in cats with chronic kidney disease at different stages, verifying whether their administration would lead to immediate side effects, such as severe clinical or laboratory abnormalities, as well as the formation of cell masses or histological changes compatible with tumor proliferation.”
Additionally, MSCs already have a well-documented safety profile in feline models and other species (lines: 390–391). Therefore, we considered that including such a group at this early stage could have masked potential adverse responses specific to fetal renal cells, such as immunogenicity, acute toxicity, or tumorigenic potential—parameters that were, in fact, the primary focus of this investigation.
Thus, we reiterate that the experimental design was structured to isolate and accurately assess the behavior of fetal renal progenitor cells—an essential step for validating their safety before proceeding to therapeutic comparisons with other established approaches, such as MSCs.
Comment #4
The study's evaluation of clinical efficacy relies heavily on subjective assessments from cat owners, including observations of increased appetite and improved activity. While these outcomes may reflect meaningful clinical improvement, they are inherently vulnerable to bias, especially given the small sample size. To enhance objectivity and reproducibility, future studies should incorporate quantifiable physiological and biochemical markers, such as longitudinal SDMA levels, glomerular filtration rate, and urinary protein concentrations.
Response
We appreciate the reviewer’s thoughtful observation regarding the subjective nature of some of the clinical evaluations described in the study. Indeed, we acknowledge that the owners’ reports—such as improvements in appetite, vitality, and social interaction—are inherently subject to bias and do not replace objective clinical parameters. However, it is important to emphasize that, at this preliminary stage of investigation, such observations were used in a complementary manner, aiming to capture early and visible clinical signs of safety and well-being following the administration of renal progenitor cells (lines 464–465).
As stated in the manuscript, “the primary focus of this study was to assess the initial safety of fetal renal cell administration, rather than to measure efficacy with statistical rigor or through multiple quantitative biomarkers.” For this reason, alongside owner-reported outcomes, objective laboratory parameters such as creatinine, urea, and complete blood count were monitored. These data were intended to support, albeit in a limited way, the absence of immediate deleterious effects (lines 462; 463–464; 467–469).
We fully agree that more objective and robust assessments—such as longitudinal SDMA measurement, glomerular filtration rate (GFR) estimation, and urinary protein quantification—are essential for future studies, particularly to enable precise evaluation of clinical efficacy. This methodological refinement is already planned as a subsequent phase following this pilot study, contingent upon the confirmation of biosafety—which, as demonstrated, has been achieved.
Therefore, we reiterate that the primary objective of this work was to ensure the absence of relevant adverse effects following the application of renal progenitor cells in cats with chronic kidney injury at varying stages, thereby laying the foundation for future investigations with more robust designs focused on quantifiable therapeutic efficacy.
Comment #5
No statistically significant differences were observed between the treatment and control groups in blood chemistry markers such as creatinine, phosphate, BUN, or SDMA. This is particularly concerning, as SDMA is recognized as an early and sensitive biomarker of renal dysfunction. The lack of change in these quantitative indicators suggests that the cell therapy may not have exerted a measurable effect on CKD pathophysiology. Without demonstrable functional improvement, a claim of safety alone is insufficient to support clinical relevance.
Response
We appreciate the reviewer’s pertinent observation and understand the concern regarding the absence of statistically significant differences in classical biochemical biomarkers such as creatinine, urea, phosphate, and SDMA. Indeed, these markers are widely used in veterinary clinical practice to monitor renal function, with SDMA standing out for its higher sensitivity in the early detection of renal dysfunction.
However, it is important to emphasize that the primary objective of this study was not to assess the functional efficacy of cell therapy in reversing or stabilizing chronic kidney disease (CKD), but rather to evaluate the biosafety of fetal renal progenitor cell administration in cats with CKD, as described in the objectives of the manuscript. (lines 91–93), “This work consisted of a pilot investigation to determine whether the administration of fetal renal progenitor cells in cats with chronic kidney disease would be clinically, laboratorially, and histologically safe, with an emphasis on the absence of immediate adverse effects and the non-formation of tumoral structures or acute rejection.”
Moreover, it is worth noting that the therapeutic group consisted of a small number of animals at varying stages of CKD (lines 227–228). This clinical heterogeneity, while relevant for observing the absence of immediate adverse effects across multiple disease stages, may also have contributed to the lack of statistically significant differences in biochemical biomarkers, as physiological responses tend to vary with disease severity. This limitation was addressed in the discussion section of the manuscript, as follows (lines 467–472):
“In this research, SDMA showed no statistically significant differences (p ≤ 0.005) between the treated and control groups. Due to the small sample size and heterogeneous renal disease stages in the treated groups, this study primarily demonstrated the safety of renal progenitor cell administration, with no side effects after application. However, further studies with larger sample sizes and more homogeneous groups of CKD should be conducted to confirm these findings.”
Therefore, we emphasize that the lack of measurable functional improvement at this stage does not compromise the aims of the investigation, which focused on evaluating the safety of cell transplantation. The clinical relevance and regenerative potential of fetal renal progenitor cells will be assessed more rigorously in future studies, which will include longer follow-up periods, expanded sample sizes, and more homogeneous clinical groups.
Comment #6
The study identifies renal progenitor cells based on the expression of markers such as CD117, WT1, and NEPHRIN. However, it does not evaluate widely accepted progenitor cell markers like SIX2 and CITED1, which are standard in human and murine renal research. Even if antibody limitations in cats present a challenge, alternative approaches such as gene expression profiling should have been pursued to strengthen the identification and characterization of the cell population.
Response
We appreciate the valuable feedback regarding the phenotypic characterization of the renal progenitor cells used in this study. However, as noted in the manuscript, we did not employ markers specific to embryonic kidney progenitor cells, as suggested by the reviewer, due to challenges in expanding the panel of renal progenitor-specific markers—particularly for the feline species. The antibodies mentioned by the reviewer are indeed highly relevant, and we have included an additional paragraph in the Discussion section highlighting their importance and explaining the reasons for not using them in our study (lines 428–436).
The antibodies used to characterize the renal progenitor cells—namely Nephrin, WT1, and CD117—were selected based on evidence from the literature. Several studies have demonstrated that Nephrin has been used to investigate metanephric lineages, particularly in kidney organoids derived from pluripotent stem cells. Authors such as Pecksen et al. (2024), Ohmori et al. (2021), Tanigawa et al. (2018), and Nishinakamura et al. (2017) have demonstrated the relevance of Nephrin, a protein essential for maintaining the filtration barrier, in studies involving organoids derived from induced pluripotent stem cells. These studies highlight the presence of this protein during nephron progenitor stages, underscoring its importance in renal development.
Regarding the possibility of conducting a gene expression profiling, we fully acknowledge that this would be an excellent approach to further support and strengthen the cellular characterization, particularly in the face of the limited availability of commercial antibodies specific to feline antigens. However, the high cost of such experiments exceeded the budget allocated for this pilot project, making their implementation unfeasible at this stage. Moreover, the scarcity of feline-specific gene data also posed a challenge, as much of the available information in the literature pertains to human and murine models. As highlighted by Lindström et al. (2018), many of these markers are conserved, suggesting that our findings may still be relevant to the feline species. Nonetheless, we emphasize that financial constraints were the primary limiting factor preventing the inclusion of more in-depth genetic analyses, despite our full awareness of their scientific value and our intention to incorporate them in future phases of this study, pending available resources.
References supporting the choice of markers used in this study:
Pecksen, E., Tkachuk, S., Schröder, C., Vives Enrich, M., Neog, A., Johnson, C. P., ... & Kiyan, Y. (2024). Monocytes prevent apoptosis of iPSCs and promote differentiation of kidney organoids. Stem Cell Research & Therapy, 15(1), 132.
Ohmori, T., De, S., Tanigawa, S., Miike, K., Islam, M., Soga, M., ... & Nishinakamura, R. (2021). Impaired NEPHRIN localization in kidney organoids derived from nephrotic patient iPS cells. Scientific Reports, 11(1), 3982.
Tanigawa, S., Islam, M., Sharmin, S., Naganuma, H., Yoshimura, Y., Haque, F., ... & Nishinakamura, R. (2018). Organoids from nephrotic disease-derived iPSCs identify impaired NEPHRIN localization and slit diaphragm formation in kidney podocytes. Stem Cell Reports, 11(3), 727–740.
Nishinakamura, R., Sharmin, S., & Taguchi, A. (2017). Induction of nephron progenitors and glomeruli from human pluripotent stem cells. Pediatric Nephrology, 32, 195–200.
WT1 is also well established in the literature as a significant marker for renal progenitor cells. As noted in a review by Kreidberg (2010), “WT1 is now known to have an important role in kidney progenitor cells during development.”
Kreidberg, J. A. (2010). WT1 and kidney progenitor cells. Organogenesis, 6(2), 61–70.
Finally, the selection of CD117 was also supported by the literature, which identifies this molecule as a marker for renal progenitor cells, as shown by Rangel et al. (2018), Rusu et al. (2018), and Patschan et al. (2006).
Rangel, E. B., Gomes, S. A., Kanashiro-Takeuchi, R., Saltzman, R. G., Wei, C., Ruiz, P., ... & Hare, J. M. (2018). Kidney-derived c-kit+ progenitor/stem cells contribute to podocyte recovery in a model of acute proteinuria. Scientific Reports, 8(1), 14723.
Rusu, M. C., Mogoantă, L., Pop, F., & Dobra, M. A. (2018). Molecular phenotypes of the human kidney: Myoid stromal cells/telocytes and myoepithelial cells. Annals of Anatomy – Anatomischer Anzeiger, 218, 95–104.
Patschan, D., Krupincza, K., Patschan, S., Zhang, Z., Hamby, C., & Goligorsky, M. S. (2006). Dynamics of mobilization and homing of endothelial progenitor cells after acute renal ischemia: modulation by ischemic preconditioning. American Journal of Physiology – Renal Physiology, 291(1), F176–F185.
Lindström, N. O., Tran, T., Guo, J., Rutledge, E., Parvez, R. K., Thornton, M. E., ... & McMahon, A. P. (2018). Conserved and divergent molecular and anatomic features of human and mouse nephron patterning. Journal of the American Society of Nephrology, 29(3), 825–840.
Comment #7
Despite CKD being a progressive, long-term condition, the study's observation period was limited to just 14 days. Such a short timeframe is inadequate for assessing sustained improvements in kidney function or the long-term impact of treatment on disease progression. The appropriateness of the CKD model itself is questionable under these constraints, and a study duration of at least several months would be necessary to evaluate therapeutic durability.
Response
Thank you for your valuable feedback. We fully acknowledge that Chronic Kidney Disease (CKD) is a progressive condition that requires long-term monitoring to accurately assess sustained therapeutic effects. In the present study, the observation period was limited to 14 days—a relatively short timeframe to measure lasting improvements in renal function or to evaluate the long-term impact of treatment on disease progression.
This temporal limitation reflects the initial and exploratory nature of the study, whose primary objective was to evaluate the immediate safety of fetal renal progenitor cells, including the absence of acute adverse effects and the assessment of tumorigenic potential following cell administration (lines 323–326; 367–373). Accordingly, a shorter observation period was selected to detect potential acute reactions and early adverse events (lines 454–455).
We fully agree that, for a robust evaluation of therapeutic efficacy and treatment durability in CKD models, follow-up studies with extended durations—ideally spanning several months—will be essential. Such studies will allow for more detailed monitoring of disease progression and renal functional response over time (lines 469–473).
Comment #8
The study utilized intraperitoneal injection for cell delivery, justifying this choice by citing ease of administration and reduced animal discomfort. However, the scientific rationale for this route is unclear. If renal homing is a desired outcome, intravenous or intra-arterial delivery is more common and supported by prior studies demonstrating superior renal engraftment. The authors also fail to provide evidence that the administered cells reached the kidney after IP injection, leaving the actual mechanism of therapeutic action uncertain.
Response
We thank the reviewer for their valuable comments. Regarding the question about the intraperitoneal route of administration, our choice was based on the fact that the linea alba is an access site devoid of major blood vessels and significant innervation. These anatomical features help reduce patient discomfort and eliminate the need for sedation. In contrast, the perirenal region carries a higher risk of direct contact with the kidney and requires administration of analgesia or sedation due to its greater sensitivity and anatomical complexity — this information has been added to the manuscript (lines 460–461).
Additionally, we considered that direct application into the renal artery would require general anesthesia and open-field surgery, substantially increasing procedural complexity and the risks to the patient. Although intravenous administration is widely used, it presents embolism formation as a major concern.
Our study was also based on previous experimental models in rats, in which the intraperitoneal route demonstrated efficacy (Barros et al., 2015; Bazhanov et al., 2016). After administration, labeled cells were successfully identified in the kidneys, supporting the suitability of this route. The references substantiating this decision have been included in the Discussion section (lines 459–461), allowing reviewers and readers to better understand the rationale for this choice.
References cited in the Discussion supporting the use of the intraperitoneal route:
Barros, M. A., Martins, J. F. P., Maria, D. A., Wenceslau, C. V., De Souza, D. M., Kerkis, A., ... & Kerkis, I. (2015). Immature dental pulp stem cells exhibited renotropic properties and pericyte-like behavior in acute kidney injury in rats. Medicina Celular, 7(3).
Bazhanov, N., Ylostalo, J. H., Bartosh, T. J., Tiblow, A., Mohammadipoor, A., Foskett, A., & Prockop, D. J. (2016). Intraperitoneally infused human mesenchymal stem cells form aggregates with mouse immune cells and attach to peritoneal organs. Stem cell research & therapy, 7(1), 27.
Geddes, R. F., Finch, N. C., Syme, H. M., & Elliott, J. (2013). The Role of Phosphorus in the Pathophysiology of Chronic Kidney Disease. Journal of Veterinary Emergency and Critical Care, 23, 122–133. https://doi.org/10.1111/vec.12032
Braff, J., Obare, E., Yerramilli, M., Elliott, J., & Yerramilli, M. (2014). Relationship between Serum Symmetric Dimethylarginine Concentration and Glomerular Filtration Rate in Cats. Journal of Veterinary Internal Medicine, 28, 1699–1701. https://doi.org/10.1111/jvim.12446
Holt, D., & Agnello, K. A. (2013). Peritoneum. Elsevier Ltd. ISBN: 9780702043369
Choi, G. J., Kang, H., Baek, C. W., Jung, Y. H., & Kim, D. R. (2015). Effect of Intraperitoneal Local Anesthetic on Pain Characteristics after Laparoscopic Cholecystectomy. World Journal of Gastroenterology, 21, 13386
I, the corresponding author of the manuscript "Development and Characterization of a Novel Lineage of Renal Progenitor Cells for Potential Use in Feline Chronic Kidney Disease: A Preliminary Study" under the assigned ID cells-3779232, on behalf of my coauthors, once again extend my heartfelt gratitude to the knowledgeable Editor-in-Chief and reviewers for their time and expertise in revising our manuscript. After we addressed their constructive and refined feedback and suggestions, a significantly improved manuscript version emerged. Undoubtedly, their insightful suggestions and feedback have significantly enhanced the quality of our manuscript. We respectfully are at the disposal of the Editor-in-Chief and the Reviewer to address any additional suggestions regarding our publication. Suppose you are satisfied with our newly refined and significantly improved version. In that case, we look forward to the acceptance of our article for publication in this prestigious journal, Cells. Thank you once again for your time and expertise.
Reviewer 2 Report
Comments and Suggestions for Authors
Investigating cell therapy is crucial for advancing treatments for chronic illnesses. This is particularly relevant for chronic kidney disease (CKD), which ranks as one of the top chronic non-communicable diseases in both humans and cats. The current research does not have a sufficiently large sample size to make more definitive conclusions. Additionally, the lack of statistical significance in all biochemical parameters may not indicate the strength of the designed study. Nonetheless, the cats that received treatment did not experience any adverse effects following the administration of renal progenitor cells. The use of the intraperitoneal route allowed for the delivery of larger quantities of cell therapy, enhancing systemic absorption and reducing the likelihood of thrombus development. In conclusion, a highly significant result of this study must not be overlooked - the application of this innovative progenitor cell lineage demonstrated no adverse effects, suggesting its safety for use in felines suffering from renal disease. Consequently, I recommend that this research be published, as it paves the way for new avenues in the management of chronic kidney disease (CKD), ensuring a safe outcome.
Author Response
RESPONSE TO REVIEWERS' COMMENTS
Manuscript number: cells-3779232 ― Cells (MDPI)
"Development and Characterization of a Novel Lineage of Renal Progenitor Cells for Potential Use in Feline Chronic Kidney Disease: A Preliminary Study"
The authors of this document wish to express their deepest gratitude to the Editor-in-Chief and the Reviewer for their thorough and insightful evaluation of our manuscript. Their expert feedback has been invaluable in enhancing the quality of our work. We have carefully considered and diligently implemented each suggestion, significantly improving the manuscript. We have made substantial revisions to address the points raised. These noteworthy changes are marked mainly with YELLOW-highlighted text throughout the document for ease of reference. A note will be provided for the referee's attention for corrections highlighted in a different color. Additionally, we have prepared a detailed and comprehensive response to each comment and suggestion. This response is organized in a "point-by-point" format below, ensuring that every concern has been thoroughly addressed and explained. We sincerely appreciate the time and effort invested by the Editor-in-Chief and the Reviewer, and we believe their contributions have significantly strengthened the final version of our manuscript.
Reviewer #2
Comment #2
Investigating cell therapy is crucial for advancing treatments for chronic illnesses. This is particularly relevant for chronic kidney disease (CKD), which ranks as one of the top chronic non-communicable diseases in both humans and cats. The current research does not have a sufficiently large sample size to make more definitive conclusions. Additionally, the lack of statistical significance in all biochemical parameters may not indicate the strength of the designed study. Nonetheless, the cats that received treatment did not experience any adverse effects following the administration of renal progenitor cells. The use of the intraperitoneal route allowed for the delivery of larger quantities of cell therapy, enhancing systemic absorption and reducing the likelihood of thrombus development. In conclusion, a highly significant result of this study must not be overlooked - the application of this innovative progenitor cell lineage demonstrated no adverse effects, suggesting its safety for use in felines suffering from renal disease. Consequently, I recommend that this research be published, as it paves the way for new avenues in the management of chronic kidney disease (CKD), ensuring a safe outcome.
Response
We would like to sincerely thank the reviewer for their thoughtful and encouraging comments. We are deeply grateful for your recognition of the relevance of our study and your appreciation of the potential of renal progenitor cell therapy in the treatment of chronic kidney disease. Your positive and constructive feedback is highly motivating and reinforces the importance of continuing to explore innovative and safe therapeutic approaches in veterinary medicine. Thank you once again for your valuable contribution to the improvement and recognition of our work.
I, the corresponding author of the manuscript "Development and Characterization of a Novel Lineage of Renal Progenitor Cells for Potential Use in Feline Chronic Kidney Disease: A Preliminary Study" under the assigned ID cells-3779232, on behalf of my coauthors, once again extend my heartfelt gratitude to the knowledgeable Editor-in-Chief and reviewers for their time and expertise in revising our manuscript. After we addressed their constructive and refined feedback and suggestions, a significantly improved manuscript version emerged. Undoubtedly, their insightful suggestions and feedback have significantly enhanced the quality of our manuscript. We respectfully are at the disposal of the Editor-in-Chief and the Reviewer to address any additional suggestions regarding our publication. Suppose you are satisfied with our newly refined and significantly improved version. In that case, we look forward to the acceptance of our article for publication in this prestigious journal, Cells. Thank you once again for your time and expertise.
Round 2
Reviewer 1 Report
Comments and Suggestions for Authors
Dear Authors,
I am satisfied with the revisions that have been made by the authors.
Author Response
Dear Erudite Reviewer, thank you for your comment and brilliant suggestions.
We appreciate your cooperation.
With best regards,
The Authors.